# Update on the Epidemiology, Diagnosis, and Treatment of Coccidioidomycosis

**DOI:** 10.3390/jof8070666

**Published:** 2022-06-25

**Authors:** Samantha L. Williams, Tom Chiller

**Affiliations:** Mycotic Diseases Branch, Centers for Disease Control and Prevention, Atlanta, GA 30329, USA; tnc3@cdc.gov

**Keywords:** coccidioidomycosis, *Coccidioides*, Valley fever, endemic mycoses, fungal diseases

## Abstract

Coccidioidomycosis is a fungal infection caused by *Coccidioides immitis* and *Coccidioides posadasii*. The dimorphic fungi live in the soils of arid and semi-arid regions of the western United States, as well as parts of Mexico, Central America, and South America. Incidence of disease has risen consistently in recent years, and the geographic distribution of *Coccidioides* spp. appears to be expanding beyond previously known areas of endemicity. Climate factors are predicted to further extend the range of environments suitable for the growth and dispersal of *Coccidioides* species. Most infections are asymptomatic, though a small proportion result in severe or life-threatening forms of disease. Primary pulmonary coccidioidomycosis is commonly mistaken for community-acquired pneumonia, often leading to inappropriate antibacterial treatment and unnecessary healthcare costs. Diagnosis of coccidioidomycosis is challenging and often relies on clinician suspicion to pursue laboratory testing. Advancements in diagnostic tools and antifungal therapy developments seek to improve the early detection and effective management of infection. This review will highlight recent updates and summarize the current understanding of the epidemiology, diagnosis, and treatment of coccidioidomycosis.

## 1. Introduction

Coccidioidomycosis, also known as Valley fever, is an infection caused by the inhalation of airborne arthroconidia from the soil-dwelling fungi, *Coccidioides* spp. Though often considered a rare disease, the environmental mycosis is a growing public health concern due to rising case counts and evidence of geographic expansion. Symptoms develop in approximately 40% of cases, frequently resembling other respiratory illnesses with signs such as cough, fever, shortness of breath, and fatigue [1]. Clinical findings may be indistinguishable from community-acquired pneumonia (CAP), which can lead to misdiagnosis and delays in appropriate antifungal treatment [2]. Although the infection is usually self-limiting, many patients require antifungal treatment to resolve illness, and a small subset of infections result in life-threatening severe pulmonary or disseminated disease [3,4]. Direct medical costs combined with potential productivity losses constitute a substantial economic burden [5,6,7,8].

In the United States, coccidioidomycosis is known to be endemic in the Southwest, with southern Arizona and the San Joaquin Valley in California comprising hyperendemic zones [9]. *Coccidioides immitis* has been found as far north as Washington state [10,11]. Globally, a combination of case reports and skin test studies also established areas of endemicity in Mexico, parts of South America (Argentina, Bolivia, Brazil, Colombia, Paraguay, Venezuela), and parts of Central America (Guatemala, Honduras), though much remains unknown about coccidioidomycosis in these regions because of limited diagnostic capabilities and reporting [12,13]. The global geographic distribution of *Coccidioides* spp. is displayed in Figure 1. Reported cases of coccidioidomycosis increased steadily in recent years and likely underestimate the true burden of disease. 

The development of more rapid diagnostics is key to early and accurate diagnosis. Despite improvements in test offerings and performance over time, challenges to coccidioidomycosis diagnosis persist. Results can be difficult to interpret, and sensitivity and specificity may vary based on immunosuppression status and stage of disease. These complexities are compounded by low clinician awareness, particularly outside of known endemic regions [14,15]. Continued developments in antifungal therapy aim to improve patient outcomes and address concerns of toxicities, though questions regarding early treatment and optimal management persist. 

This review summarizes the current understanding of the epidemiology, diagnosis, and treatment and management of coccidioidomycosis. 

## 2. Epidemiology

### 2.1. Increased Number of Reported Cases

Coccidioidomycosis, caused by *Coccidioides immitis* and *Coccidioides posadasii*, is a nationally notifiable disease in the United States, though it is reportable only in 26 states and the District of Columbia [16]. Reportable status is designated by the state or jurisdiction and requires healthcare professionals and laboratories to notify public health departments of cases that meet the Council of State and Territorial Epidemiologists (CSTE) definition. For nationally notifiable diseases or conditions, states voluntarily submit case data to the US Centers for Disease Control and Prevention (CDC) for patients meeting CSTE criteria. The number of cases reported to the CDC rose considerably since 2014, as shown in Figure 2. Following a three-year decline from 2012–2014, case counts more than doubled from 8232 in 2014 to 20,003 in 2019 [17]. Arizona and California, which account for more than 95% of reported cases, showed similar trends in rates of disease. Incidence in Arizona grew from 84.4 cases per 100,000 population to 144.1 per 100,000 from 2014–2019, while California’s incidence more than tripled from 6.0 per 100,000 population to 22.5 per 100,000 population in the same timeframe [18,19]. Based on provisional counts, Arizona’s incidence increased to 161.1 per 100,000 population in 2020 and stayed relatively even at 159.8 per 100,000 population in 2021, while incidence in California dipped to 16.9 per 100,000 in 2020 before rising to 20.7 per 100,000 in 2021 [20,21].

The cause of the concerning increase is likely multifaceted. Environmental factors favorable to the growth and subsequent dispersal of *Coccidioides* spp. may have contributed to a higher frequency of disease [22,23]. The causative fungus is known to live in arid and semi-arid regions, and statistical models demonstrated the influence of temperature and precipitation patterns on the proliferation of *Coccidioides* spp., though only a weak correlation was found in some highly endemic areas of California [22,24,25,26,27,28,29]. Periods of precipitation facilitate fungal growth in the environment, while ensuing periods of low precipitation and high temperature create ideal conditions to release fungal spores [28,29,30,31]. These effects may be amplified after droughts; incidence in California declined during the 2007–2009 and 2012–2015 droughts but increased markedly in the following two years [23]. Particulate matter of size less than 10 µm (PM10) is also thought to impact coccidioidomycosis incidence, and PM10 concentration rose by 12% in the Southwest and 29% in the West from 2010–2020 [30,32,33].

Population growth may have impacted the rising coccidioidomycosis case counts. Arizona continues to be one of the fastest-growing states, recording a 12% population increase from 2010 to 2020, driven primarily by a 16% increase in the highly-populated Maricopa County; California also experienced a 6% population increase during the same timeframe [34]. Many of the incoming residents are likely immunologically naïve to coccidioidomycosis. Across several endemic regions, desert land was converted to urban and suburban centers to accommodate population influx, resulting in substantial soil disturbance and potential exposure [35,36].

Advances in healthcare practices have expanded the at-risk population for coccidioidomycosis. Prolonged life spans have led to a growing population over 65 years of age [34]. This group has a higher prevalence of chronic disease and is more frequently diagnosed with *Coccidioides* spp. infection [37]. Developments in therapeutics and medical procedures extended survival for patients with weakened immune systems or previously fatal conditions. The number of stem cell transplants increased by 8% from 2015–2019, while solid organ transplantation grew by 45% from 2011–2021 [38,39]. Use of immunosuppressive agents has become more widespread with greater availability [40]. Transplants and immunosuppressants both represent known risk factors for developing severe coccidioidomycosis [41,42].

Additional factors that may have contributed to the increase in coccidioidomycosis case counts include changed or improved reporting practices, increased laboratory testing, or heightened awareness. Patients in Arizona who heard of the disease before seeking healthcare were diagnosed earlier than patients who were unaware of coccidioidomycosis and were also more likely to request testing [43]. However, surveys of representative samples showed low Valley fever awareness, even in regions of known endemicity [44,45]. In California, only 25.0% of respondents living in high-incidence areas knew that *Coccidioides* spp. (termed ‘Valley fever fungus’ in the survey) existed in their area of residence, and just 3.5% of respondents with a risk factor for severe coccidioidomycosis knew that they were at an elevated risk for severe infection [45]. Tailored messaging to vulnerable populations may increase the knowledge of disease and consequently influence healthcare-seeking behavior and testing practices.

### 2.2. Geographic Expansion of Coccidioides Species

The initial geographic distribution of coccidioidomycosis was established in the 1940s and 1950s through extensive coccidioidin skin tests to assess prevalence [9]. Although many cases reported outside of the traditional areas of endemicity are attributed to travel in endemic regions or reactivation of a latent infection, several outbreaks in California and Utah have indicated that the geographic range is extending northward [46,47,48]. Whole-genome sequencing confirmed the local acquisition of coccidioidomycosis in 2010 in Washington. A clinical isolate from the patient and soil isolates retrieved from the suspected point of exposure were found to be identical, providing the first evidence of *Coccidioides* endemicity in the state [10,49].

Reasons for the expansion beyond traditionally recognized areas are not certain, though several have been theorized. Climate factors are known to influence environmental dynamics, and it is hypothesized they may create suitable conditions for *Coccidioides* spp. habitation in areas that did not previously support fungal growth. A climate niche model used disease incidence data and climate projections to predict that, by the year 2100, the area of coccidioidomycosis endemicity will more than double and cases will increase by 50%, based on global warming scenarios [24]. Although evidence demonstrated the potential influence of climate change on the future geographic distribution of *Coccidioides* spp., it is unclear whether climate contributed to the current observed expansion. Climate change may additionally trigger an increase in severe weather events, such as wildfires, which have been linked to infection and cause substantial disruption and dispersal of soil [50,51]. Dust storms (haboobs) were previously thought to drive increases in coccidioidomycosis incidence, but recent studies found no significant association [52,53,54]. Air sampling studies in Arizona even suggest that dust storms may diminish the concentration of arthroconidia in the air [54].

Some research also suggests that rodents may contribute to the geographic distribution by serving as a reservoir for coccidioidomycosis, though conflicting results called this theory into question. The hypothesis, first proposed in the 1940s, was largely discarded following contradictory results from large-scale soil sampling [55,56]. However, several studies since found a correlation between the amount of *Coccidioides* spp. recovered from soil obtained from rodent burrows compared with surrounding topsoil [57,58,59]. Contemporary genomic analysis also spurred a resurgence of the theory, as the findings indicated a higher proportion of animal versus plant tissue-associated genes [60]. Notably, recent systematic soil sampling in Washington did not yield any association between rodent burrows and the presence of *Coccidioides* [49]. The rodent hypothesis offers a conceivable explanation for the fungus’ patchy distribution in endemic regions, but more research is needed to better understand the role of rodents in the *Coccidioides* life cycle. Continued surveillance will be essential to monitor trends in case counts and geographic expansion. 

### 2.3. Risk Factors

Coccidioidomycosis can affect anyone who is exposed to the causative fungus, but certain groups may be at higher risk of infection or severe disease. People with weakened immune systems have demonstrated greater susceptibility to disease. Elevated risk exists for severe coccidioidomycosis among people living with HIV/AIDS, particularly those with low CD4 counts, though incidence among this population decreased with the advent of antiretroviral therapy [61,62,63]. Transplant recipients and patients receiving immunosuppressive medications such as corticosteroids, chemotherapy, or tumor necrosis factor inhibitors represent additional at-risk groups owing to cellular immunodeficiencies [64,65]. 

Reactivation of a latent infection is also a concern among transplant recipients. Transplant centers in endemic areas may administer antifungal prophylaxis to prevent recurrence of coccidioidomycosis, often up to one-year post-transplantation [4]. Both universal and targeted programs aimed at patients with positive serologic results or a history of *Coccidioides* spp. infection demonstrated encouraging results [66,67,68,69]. However, no consensus exists regarding drug type or duration [69,70]. Further research is needed to define optimal prophylaxis strategies among certain patient groups. 

Pregnancy is an established risk factor for coccidioidomycosis. Evidence suggests a correlation between the severity of coccidioidomycosis and the length of gestation, with the most severe disease occurring in the late stages of pregnancy and the immediate post-partum period. Illness in pregnant women is further complicated by treatment challenges because azole antifungals are known teratogens [71,72]. Epidemiologic studies showed that people of African American or Filipino descent are at a heightened risk for infection, particularly severe or disseminated disease, though reasons are unknown [73,74,75].

Occupational hazards have been documented for workers exposed to dust from soil disturbance. Jobs involving digging, excavation, or soil disruption in endemic areas are considered to pose the greatest risk of coccidioidomycosis; common examples include construction, archaeology, agriculture, firefighting, and mining, gas, or oil extraction [76]. Although the aforementioned activities are more commonly associated with *Coccidioides* spp. exposure, cases related to minimal soil disturbance have been reported, such as an outbreak among cast and crew on an outdoor television set, as well as extremely high rates among employees and inmates at state prisons in California [77,78]. 

Researchers have long postulated that males are at an elevated risk of *Coccidioides* spp. infection based on the observed differences in disease incidence by sex [79,80,81]. Some claim that the disparity is likely because of the disproportionate representation of males in occupations or recreational activities associated with dust or soil exposure. Recent data furthered the assertion of sex as an independent risk factor through a survey of human patients, veterinary patients, and nonhuman primates [82]. Results showed significantly higher rates of infection, severe disease, and greater average maximum serum complement fixation (CF) titers among human males compared with females. Significant differences in incidence emerged at age 19 and remained until age 80. Additionally, nonhuman primate and unaltered canine males exhibited a significantly increased risk for coccidioidomycosis compared with their female counterparts, though no significant sex differences were observed between castrated male dogs and female dogs [82]. Findings from this study suggest that biological mechanisms may contribute to the reported differences in *Coccidioides* spp. infection between males and females.

### 2.4. Coccidioidomycosis and COVID-19

Several reports described a co-infection of *Coccidioides* spp. and severe acute respiratory syndrome novel coronavirus (SARS-CoV-2), the causative agent of coronavirus disease 2019 (COVID-19) [83,84,85,86,87,88,89,90]. Common risk factors include older age, diabetes, immunosuppression, African American or Latino heritage, and smoking [91]. The interaction between the two infections remains largely unclear. Underlying respiratory illness and chronic lung disease are associated with severe COVID-19, suggesting that patients with chronic pulmonary coccidioidomycosis may be at an elevated risk of severe COVID-19 [92,93,94]. SARS-CoV-2 infection may also increase the risk of coccidioidomycosis reactivation because of immune dysregulation within the host, though at present only one case report is indicative of reactivation [90,95,96]. 

The COVID-19 pandemic may have lengthened delays in diagnosis as a result of similar symptoms between SARS-CoV-2 and *Coccidioides* spp. infection. A survey of infectious disease physicians indicated that testing practices did not change because of COVID-19, though primary care doctors were not among the respondents to this survey [97]. It is also unknown whether health-seeking behavior or access to healthcare during the pandemic affected coccidioidomycosis case counts. Further research is needed to understand the relationship between coccidioidomycosis and COVID-19.

## 3. Diagnosis

### 3.1. Diagnostic Challenges

Healthcare providers are faced with a variety of challenges in relation to coccidioidomycosis diagnosis, though they generally fall into two categories: (1) technical efficiency of laboratory testing and (2) provider knowledge and behavior. Nonspecific symptoms resemble other respiratory illnesses, and laboratory test results may be difficult to interpret or logistically challenging. A delayed or missed diagnosis can consequently lead to adverse patient outcomes in the absence of proper antifungal treatment. Presenting symptoms are often mistaken for bacterial or viral pneumonia, yet current Infectious Disease Society of America (IDSA) diagnostic guidelines for CAP do not recommend testing for *Coccidioides* spp. infection, despite evidence that the fungal disease accounts for up to one third of CAP etiologies in some endemic areas [98,99].

According to recent enhanced surveillance, 70% of patients had another condition diagnosed before being tested for coccidioidomycosis, and the median duration from seeking healthcare to diagnosis was 38 days [14]. Another study found that 70% of patients received antibiotics in the three months before their first positive coccidioidal test, and those patients were prescribed a median of three antibiotic courses [100]. Implications of inappropriate antibacterial treatment can include drug resistance and unnecessary costs. 

A variety of laboratory diagnostic tests are available to detect *Coccidioides* spp. infection, but testing rates remain low. Performance measures and considerations for the various laboratory tests are described in Table 1. A survey of healthcare providers revealed that only 3.7% reported “frequently” testing CAP patients for coccidioidomycosis, and 15.0% tested “sometimes”. Even in Arizona and California, just 32.4% and 7.4% of providers, respectively, reported frequent testing [15]. Just over 60% of healthcare providers surveyed in Arizona were confident in their ability to diagnose *Coccidioides* infection, and just under 60% claimed to be knowledgeable about laboratory tests used for detecting coccidioidal antibodies [101].

A recent assessment of practice patterns in an Arizona healthcare system showed that 73% of coccidioidomycosis diagnoses were made during hospital admission. Nearly half of those patients had at least one prior healthcare encounter related to their symptoms, but coccidioidal serology was only obtained for 29% of them during those visits [102]. Their subsequent hospitalizations suggest that retesting may have been beneficial to counteract any false negatives in the early stages of the disease. 

### 3.2. Serology

Serologic antibody testing is the most frequently used diagnostic tool for coccidioidomycosis. Antibody development may trail illness onset by several weeks; serial testing is therefore recommended, should symptoms persist following an initial negative test result. Consideration of other laboratory tests may be warranted for immunosuppressed patients, as sensitivities for coccidioidal antibody tests are generally lower among this population.

Enzyme immunoassay (EIA) tests are widely available and offer rapid results, detecting coccidioidal antibodies within hours. EIA testing of both immunoglobulin (Ig) M and IgG levels can be highly sensitive (59–88%) and specific (68–90%), though results are variable [103,104,105]. Results for IgG EIA alone are generally preferred to those for IgM alone, as IgM EIA tests are known to show false positives and should be interpreted with caution; IgG EIA test sensitivities range from 47–87% with specificities of 90–96%, while IgM EIA test sensitivities range from 22–61% with specificities of 70–99% [104,105,106]. A lateral flow assay (LFA) has been developed to offer a fast and simple alternative for antibody detection, but initial data have shown markedly lower sensitivity compared to EIA [107].

Alternative serologic tests for antibody detection include immunodiffusion (ID) and complement fixation (CF), which are less sensitive than EIA but more specific and commonly serve as confirmatory tests [103]. ID tests measure IgG or IgM but can take several days to return results, limiting their utility for quick diagnosis. Quantitative CF results are valuable measures to assess disease severity and progression; higher CF titers may indicate dissemination, and increasing titers are associated with clinical deterioration. However, CF tests measure only IgG antibodies, and cross-reactivity with other dimorphic fungi may influence results [108,109,110]. All coccidioidomycosis antibody tests involve specialty equipment, and CF testing also requires a high level of specialized training. Methods for ID and CF vary across laboratories, leading to a considerable variability of results. Standardization is needed to increase the reliability and comparability of these diagnostic tools. 

### 3.3. Antigen Detection

Antigen testing may inform the detection of *Coccidioides* in early stages of disease, particularly among immunocompromised patients [111,112]. A commercially-available test for serum, urine, or cerebrospinal fluid (CSF) samples has demonstrated high specificity. Sensitivity is moderate among immunocompromised populations but drops considerably among immunocompetent patients, and CSF antigen testing is sensitive only in patients with coccidioidal meningitis. Cross-reactivity with blastomycosis and histoplasmosis may complicate diagnosis in the absence of additional testing [109,111,113].

### 3.4. Microscopy and Culture

Identification of *Coccidioides* in clinical specimens by culture remains the gold standard for coccidioidomycosis diagnosis, though biosafety concerns and potential challenges visualizing the spherules may limit the use of this method. Additionally, culture growth may take up to a week, and sensitivity is dependent on specimen quality. 

Histopathology and cytopathology represent other traditional methods used to identify the organism in clinical specimens. Sensitivities of histopathology (23–84%) and cytopathology (15–75%) also vary greatly [114].

### 3.5. Additional Laboratory Diagnostic Methods

Encouraging results from polymerase chain reaction (PCR) testing demonstrate the potential value of molecular methods as a diagnostic tool for coccidioidomycosis. At present, PCR tests are not commonly used directly on clinical specimens, and the site of specimen collection may impact test performance [115,116]. A study to evaluate performance of the serum (1→3) β-d-glucan (BG) assay showed sensitivity (43.9%) and specificity (91.1%) comparable to BG testing for other invasive mycoses, such as aspergillosis and candidiasis. However, sensitivity was lower among patients with acute pulmonary coccidioidomycosis (19.1%), specificity was determined against healthy controls, and BG values correlated poorly with serum coccidioidal CF titers [117]. The utility of BG testing as a clinical diagnostic tool is limited by its inability to detect specific pathogens.

**Table 1 jof-08-00666-t001:** Performance and considerations for coccidioidomycosis laboratory diagnostic tests.

Test	Sensitivity	Specificity ^‡^	Considerations
**Serology**			
Antibody			Antibody production may lag behind symptom onset. Sensitivity is often lower in immunosuppressed patients.
*EIA IgG or IgM* [103,104,105]	59–88%	68–96%	Rapid performance time within hours. Often used as a screening test, later confirmed by ID or CF. IgM only may lead to more false positives than IgG only.
*EIA IgG* [103,104,105]	47–87%	89–97%	
*EIA IgM* [103,104,105]	22–61%	70–99%	
*ID ^§^* [103,118]	60–91%	99–100%	Results may take several days to receive. Some specialized training is required. Methods are not standardized across laboratories.
*CF ^§^* [103,108,109,118]	65–98%	80–98%	Titers may offer prognostic value of disease progression. Measurement of IgG only. Highly specialized training is required. Methods are not standardized across laboratories.
*LFA ^§^* [117,118]	31–99%	92–98%	Rapid 1-h performance time.
**Antigen**			
Urine and serum [113]	57%	99%	May detect *Coccidioides* in the early stages of the disease [112]. May be preferred to antibody tests for immunocompromised patients. Substantial cross-reactivity with other dimorphic fungi.
Urine [111,113]	37–71%	99%	
Serum [119]	73%	100%	
**Microscopy and culture**			
Culture [114]	23–93%	High	Considered the gold standard of coccidioidomycosis diagnosis. Biosafety level 3 lab needed for safe isolation of *Coccidioides*. Culture growth may take up to a week. Sensitivity is heavily dependent on specimen quality.
Histopathology [114]	23–84%	High	
Cytology [114]	15–75%	High	
**Additional laboratory methods**			
PCR [115,116]	56–75%	99–100%	Rapid 4-h performance time. Site of specimen collection may influence results.
(1→3) β-d-glucan [117]	44%	91%	Lower sensitivity among patients with acute pulmonary coccidioidomycosis. Values correlate poorly with CF titers. Test cannot detect specific pathogens.

Abbreviations: CF, complement fixation; EIA, enzyme immunoassay; ID, immunodiffusion; IgG, immunoglobulin G; IgM, immunoglobulin M; PCR, polymerase chain reaction. ^‡^ Specificity is based on published results; estimates may not be directly comparable, as different control populations were used in some cases. *^§^* Sensitivity and specificity ranges include testing from outbreak investigations.

## 4. Treatment

Management of coccidioidomycosis often depends on the severity of disease and clinical history of the patient. Acute pulmonary coccidioidomycosis in immunocompetent hosts typically resolves without antifungal intervention and results in life-long immunity [3]. Regular assessment to monitor symptoms and radiological results may prove sufficient for these patients. Some physicians advise empiric antifungal therapy to shorten symptom duration and prevent dissemination, though there are no data from prospective randomized clinical trials to support the efficacy of early treatment on these outcomes [120].

Antifungal treatment for primary pulmonary coccidioidomycosis is recommended for immunosuppressed hosts or patients with particularly devastating forms of disease. IDSA guidelines outline symptoms that may warrant the initiation of therapy; these symptoms include substantial weight loss, persistent intense night sweats, infiltrates involving more than half of 1 lung or portions of both lungs, prominent or persistent hilar adenopathy, CF titers exceeding 1:16, inability to work, or symptoms that persist for more than 2 months. Additional considerations for early treatment include a history of diabetes, frailty because of old age, comorbidities, or African American or Filipino ancestry, though the IDSA guidelines note that ethnicity and diabetes status should only modestly influence management decisions [4].

### 4.1. Azoles

The advent of azole therapy significantly influenced the antifungal treatment of coccidioidomycosis. Ketoconazole became the first azole used against *Coccidioides* in 1981, though it is no longer recommended because of concerns of adverse effects and apparently superior efficacy of other drugs [121,122]. Fluconazole is the most commonly prescribed antifungal agent for coccidioidomycosis, available in oral and intravenous formulations. Its low cost, tolerability, and penetration into most body sites make it an appealing option for drug administration [4,123]. Side effects may include alopecia, dry skin, chapped lips, and arthropathy, and effects may become more pronounced with higher dosage [124]. Fluconazole in vitro minimum inhibitory concentrations (MICs) were found to be significantly higher than those of other triazoles in limited reports, though this has not been correlated with patient outcomes [125].

Itraconazole is also commonly used to treat coccidioidomycosis and is available as an oral solution or a capsule. Findings from a randomized double-blind trial showed that itraconazole had a higher efficacy (70% response rate) compared to fluconazole (37% response rate) for patients with skeletal forms of disease, though absorption can be challenging. Relapse rates were also lower for patients treated with itraconazole (18%) as opposed to fluconazole (28%) [126]. Hypertension, hypokalemia, sodium retention, and decreased myocardial contractility constitute reported adverse effects from itraconazole therapy, and consequently the drug is not recommended for patients at risk of heart failure [127,128,129].

Voriconazole may be prescribed for patients who are intolerant or unresponsive to fluconazole or itraconazole. It can be administered intravenously or through oral formulation and has extensive distribution throughout the body, including CSF penetration. Although voriconazole has exhibited efficacy in cases of coccidioidal meningitis, concerns of drug-drug interactions and toxicities may limit its use [130,131,132]. Voriconazole has been associated with visual impairments, altered mental status, and harmful cutaneous effects, including photodermatitis, melanoma, and squamous cell carcinoma [133,134,135].

Posaconazole is an alternate antifungal option for refractory cases of coccidioidomycosis. Originally available only as an oral suspension, the development of an intravenous formulation and delayed response oral tablet markedly improved absorption [136]. Posaconazole penetrates most body sites, with the exception of CSF, and is generally considered to be effective, even demonstrating superior performance to other triazoles in murine models [130,137,138,139,140,141]. Side effects are commonly gastrointestinal in nature, including nausea, vomiting, and diarrhea. Hypokalemia, hypertension, and peripheral edema are also reported to be associated with Posaconazole treatment [98].

Most recently, isavuconazole has been made available in oral and intravenous formulations. It is widely distributed throughout the body, yet although isavuconazole has proven to be effective against other mycoses, data for coccidioidomycosis patients in a clinical setting are limited [142,143,144]. Gastrointestinal side effects similar to those of Posaconazole have been observed. 

### 4.2. Polyenes

Prior to the introduction of triazole antifungal therapy, amphotericin B served as the primary agent for coccidioidomycosis treatment. Multiple formulations are available intravenously: amphotericin B deoxycholate (AmBd), liposomal amphotericin B (L-AMB), amphotericin B colloidal dispersion (ABCD), and amphotericin B lipid complex (ABLC). All forms are associated with nephrotoxicity, though the frequency of adverse events is generally lower in the lipid variations [145,146]. Use of amphotericin B is now primarily limited to cases that are intolerant or resistant to triazoles. However, intrathecal AmBd may still be administered in patients with coccidioidal meningitis or in their first trimester of pregnancy to avoid potential teratogenic effects of triazoles [4,147].

### 4.3. Treatment Duration and Follow-Up

Duration of coccidioidomycosis treatment varies based on disease type and progression. Therapy for uncomplicated acute pulmonary infection is commonly discontinued after 3–6 months, whereas patients with severe or chronic forms of disease may require life-long treatment. Antifungal regimens should be routinely assessed for possible adverse effects, drug-drug interactions, and therapeutic drug monitoring if needed. Cessation of treatment is generally prompted by diminished symptoms, improvements of imaging results, and declining CF titers. 

Regardless of treatment status, clinical follow-up is essential to evaluate the resolution of signs and symptoms and to identify possible relapse or dissemination. Assessments often incorporate a combination of serologic testing, radiological examination, and patient interview to monitor the course of infection. Follow-up is recommended for at least one year once the patient shows signs of improvement. 

## 5. Conclusions

Our understanding of coccidioidomycosis has no doubt deepened in recent years, but much remains to be learned. Incidence of disease is rising, and the geographic range of *Coccidioides* spp. is growing. Continued and expanded surveillance is needed in the United States and across Central and South America to monitor trends and identify potential new areas of endemicity to inform public health efforts. Increased clinician awareness and knowledge of suitable diagnostic methods are essential to improve early detection and avoid inappropriate treatment and unnecessary medical costs. Furthermore, the development of new and more rapid diagnostic tools, as well as antifungal therapies that target *Coccidioides* spp., is necessary to advance the diagnosis and subsequent resolution of disease. Insights into the epidemiology, diagnosis, and treatment of coccidioidomycosis can be used to guide future prevention and management strategies to minimize morbidity and mortality from this important disease. 

## Figures and Tables

**Figure 1 jof-08-00666-f001:**
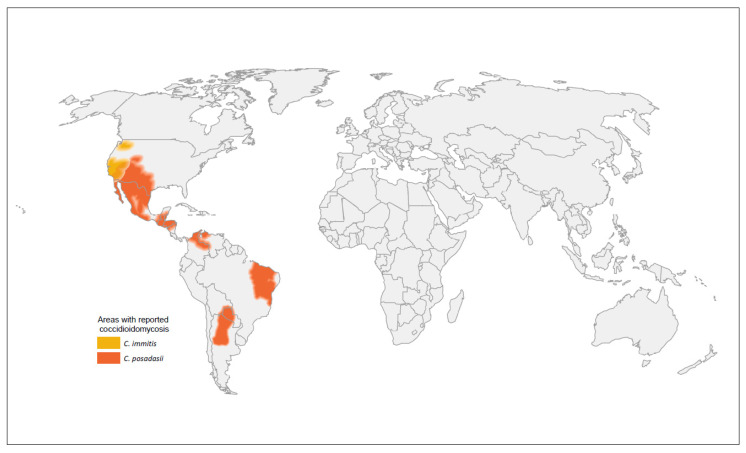
Global geographic distribution of the *Coccidioides* species.

**Figure 2 jof-08-00666-f002:**
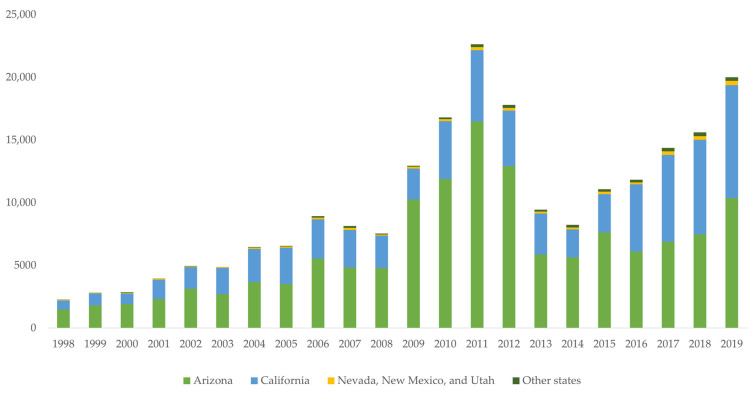
Coccidioidomycosis case counts submitted to the National Notifiable Diseases Surveillance System, 1998–2019. Case counts reported by individual states might differ slightly from those reported by the National Notifiable Diseases Surveillance System because of differences in the timing of reports or surveillance methods.

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
