# Peer review of "Update on the Epidemiology, Diagnosis, and Treatment of Coccidioidomycosis"

_jof, 2022, doi:10.3390/jof8070666_

Round 1

Reviewer 1 Report

This is a review paper regarding the epidemiology and diagnosis of coccidioidomycosis.  It covers some updated topics such as increasing range of the disease based on current environmental research and climate data modeling. It briefly discusses global impact but rapidly hones in on the impact in the American southwest as the region where the disease has the greatest medical and economic impacts. Risk factors, diagnostic tests and challenges, and treatment are broken down and discussed.

A comment on the manuscript as a whole is that the perfect past verb tense used in most places throughout is laborious and difficult to read. Seriously consider modifying most to simple past tense or using active verb tenses during revision.  This will really improve readability.  Overall, it is a well-written manuscript in terms of scope and detail.

Line 47, 79-80 – please provide references

Line 95 – consider replace with “immunologically naïve” rather than made-up word “immunonaive”

Line 93 – typo “a”

100-102 – This sentence is complex and hard to read.  Try to simplify for easier reading.

Line 108 – requires reference

Line 115 – a lot less people know “Coccidioides species” exist than “Valley Fever” since you are talking about the knowledge of the general public. If the survey tool asked whether the respondents knew about “coccidoides spp” rather than “Valley Fever,” I am sure it was quite low.  If that is what the survey tool asked about, you might want to comment on that.

The figures for this manuscript are very nice and really enhance it.

Line 183-84 – consider “. . . because azole antifungals are known teratogens.”  It is noted in the prescribing information for these drugs and is a more accurate way to state the fetal risks. The sentence is also awkward as currently written.

Line 206 – Ref 80 demonstrates that the sex differences in infection rate in dogs disappears when males are castrated.  There is a difference in rate in intact males vs intact females and all females.  This sentence needs to be modified for accuracy.  What this study strongly suggests is that the differences are probably influenced by testosterone, and this is true of some other infectious diseases as well, e.g. malaria.

3.1 – Diagnostic challenges.  Challenges appear to be related to 1) technical efficiency of testing and 2) behavior and knowledge of healthcare providers.  Consider state these concisely as a means of organizing the remainder of the section on diagnosis.

Line 283 – the ID test methods indeed “do” vary across the laboratories, not just “may.”  UC Davis serology lab definitely uses a different method, and interpretation of these tests is also made variable by operator skill, similar to CF tests.  I think  your statement that standardization is needed (lines 184-85) is the most important statement and conclusion of your section on Ab testing.

Line 289 – needs a reference. Also, I don’t think the sensitivity metrics of the Ag test are in line with antibody detection.  Please review the references again.  Urine Ag detection of cocci infection is under 33% at best and about 85-87% with Ab testing.  It is possible that you specifically meant among immunocompromised patients with severe disease but if so, please clarify.  Please separate the Ag testing from your concluding statements regarding Ab testing.

3.2 microscopy – This short section is diverse and blends in culture of clinical specimens with histopathology and cytopathology and maybe they should be at least separated into different paragraphs.  The biosafety concerns (BMBL) associated with fresh and fixed tissue specimens are significantly different than mycelial plate cultures, which yield infectious arthroconidia, a much more significant personnel hazard.

Table 1 is kind of hard to read.  The dashes in column 4 do not enhance readability of the table with format in the version being reviewed.  Some cells have a header (EIA) and others do not.  Please make the table format uniform. Antigen test – please provide a reference that says it can be detected in early stages of disease. 

Line 346 – “available” is a better term than offered to discuss drug formulations.

Line 347 – Itraconazole is an “oral solution” or a capsule.  Remove “formulation”

Line 396 – the follow up you have listed in this sentence does not assess fungal growth.  There is a very limited spectrum of patients that receive follow up fungal culture testing if the disease is clinically responding.  I suggest just remove “fungal growth.”

Author Response

Thank you for your helpful and thorough review. We updated the manuscript to reflect many of the reviewer’s suggestions. In addition to the specific line-item updates, we adjusted verb tenses where appropriate to improve the overall readability. We hope that these updates strengthen the manuscript as a whole. Responses to line-item feedback are in the attached document in red.

Reviewer 2 Report

The review of Coccidioidomycosis by S Williams and T Chiller is well written and thorough.  I have only minor comments.

Please explain further on lines 50 and 51:  “…and sensitivity and specificity may vary based 50 on medical history and disease progression.”  Sensitivity and specificity usually relate to tests, not medical history and disease progression.

Line 84:  Please change to “fungal growth in the environment, while ensuing periods”

Line 93:  Please fix typo:  “driven primarily by a 16% increase…”

Lines 268-269: Grys T et al. 2018 in Med. Mycol. have reported that sensitivities are lower than 88%.  They reported Immy and Meridian sensitivities at 53% and 64%, respectively. 

Lines 290-291:  Please mention that CSF antigen testing is sensitive only in patients with coccidioidal meningitis.  Current antigen testing is insensitive except for patients with high fungal burden.

Author Response

(The authors gave the same response as above.)
